# Ferroptosis in Haematological Malignancies and Associated Therapeutic Nanotechnologies

**DOI:** 10.3390/ijms24087661

**Published:** 2023-04-21

**Authors:** Rachel L. Mynott, Ali Habib, Oliver G. Best, Craig T. Wallington-Gates

**Affiliations:** 1Flinders Health and Medical Research Institute, College of Medicine & Public Health, Flinders University, Adelaide, SA 5042, Australia; 2Flinders Medical Centre, Bedford Park, SA 5042, Australia

**Keywords:** ferroptosis, nanotechnology, nanomedicine, haematological malignancies

## Abstract

Haematological malignancies are heterogeneous groups of cancers of the bone marrow, blood or lymph nodes, and while therapeutic advances have greatly improved the lifespan and quality of life of those afflicted, many of these cancers remain incurable. The iron-dependent, lipid oxidation-mediated form of cell death, ferroptosis, has emerged as a promising pathway to induce cancer cell death, particularly in those malignancies that are resistant to traditional apoptosis-inducing therapies. Although promising findings have been published in several solid and haematological malignancies, the major drawbacks of ferroptosis-inducing therapies are efficient drug delivery and toxicities to healthy tissue. The development of tumour-targeting and precision medicines, particularly when combined with nanotechnologies, holds potential as a way in which to overcome these obstacles and progress ferroptosis-inducing therapies into the clinic. Here, we review the current state-of-play of ferroptosis in haematological malignancies as well as encouraging discoveries in the field of ferroptosis nanotechnologies. While the research into ferroptosis nanotechnologies in haematological malignancies is limited, its pre-clinical success in solid tumours suggests this is a very feasible therapeutic approach to treat blood cancers such as multiple myeloma, lymphoma and leukaemia.

## 1. Introduction

Haematological malignancies comprise a large group of heterogeneous tumours that originate in blood-forming tissue, such as the bone marrow, or in the cells of the immune system. Broadly speaking, these tumours are grouped into those of myeloid or lymphoid origin and can be acute or chronic with regard to their natural history. Examples of bone marrow-derived malignancies include acute myeloid leukaemia, acute lymphoblastic leukaemia and chronic myeloid leukaemia, whilst chronic lymphocytic leukaemia, lymphomas and multiple myeloma are lymphoid malignancies originating outside the bone marrow in blood or lymphatic tissue. Given their biological heterogeneity, there are consequential differences in diagnostic, prognostic and therapeutic algorithms, with many being treatable but not curable.

The term ferroptosis was coined in 2012 to describe a new form of regulated cell death characterised by the iron-dependent accumulation of lipid-reactive oxygen species to lethal levels [1]. The sensitivity to ferroptosis is tightly linked to numerous biological processes, including the metabolism of amino acids, iron and polyunsaturated fatty acids, and the biosynthesis of glutathione and phospholipids. Ferroptosis has been implicated in the pathological cell death associated with degenerative diseases (e.g., Alzheimer’s disease), stroke and traumatic brain injury; however, ferroptosis may also have a tumour-suppressive function that could be harnessed for cancer therapy [2]. Importantly, lipid metabolism is intimately involved in determining cellular sensitivity to ferroptosis, with certain polyunsaturated phospholipids being susceptible to the iron-dependent lipid peroxidation necessary for its execution [3]. It is this dependence on certain phospholipids that paves the way for potential ferroptosis-inducing nanotechnologies, in particular those based on a liposome design, to be developed.

Cancer therapeutics have advanced tremendously over the past few decades, with firstly, a shift towards tumour-targeting mechanisms and, more recently, true patient-individualised or precision strategies [4,5]. An example of the former is the monoclonal antibody daratumumab, which targets CD38 on the surface of myeloma plasma cells, whilst a precision therapeutic could be the application of venetoclax in those myeloma patients whose plasma cells harbour t(11;14), which upregulates bcl-2, the target of venetoclax [6,7]. Thus, cancer drug classes have expanded to include traditional chemotherapeutics such as DNA damaging agents and mitotic spindle inhibitors, tumour-targeting small molecule inhibitors, tumour-targeting immunological agents (monoclonal antibodies, bi-specific antibodies, antibody-drug conjugates, CAR-T cells) and precision therapeutics, which may incorporate any one or more of the other therapeutic classes [4]. Nanotechnologies, on the other hand, are often seen as vehicles for existing cancer drugs in an effort to maximise cancer cell cytotoxicity through improved drug pharmacokinetics and pharmacodynamics [8]. This approach aims to not only increased tumour killing but also minimise both on-target and off-target related drug toxicities, particularly when incorporating tumour-targeting and precision principles.

In this review, we describe the major haematological cancers, ferroptosis and clinical applications thereof before providing an in-depth discourse on the use of ferroptosis-inducing nanotechnologies in cancer, focusing on those of haematological origin. We provide examples of dendritic mesoporous silica nanoparticles, iron oxide nanoparticles, micelles and liposomes, whether or not employing tumour-targeting or precision approaches. Throughout, we highlight the significance of liposome-based nanotechnologies given their importance in not only being drug carriers but also ferroptosis-inducing therapeutics in themselves by supplying relevant phospholipids to the cancer cell.

## 2. Multiple Myeloma

Multiple myeloma (MM) is a malignancy of antibody-producing plasma cells of the bone marrow [9]. In Australia, in 2018, 2247 people were diagnosed with MM at a median age of 71 years, with the overall survival at 5 years being 55% [10]. A diagnosis of MM is defined as the presence of more than 10% clonal plasma cells in the bone marrow, as well as one or more of the “CRAB” criteria: hyper**c**alcaemia, **r**enal insufficiency, **a**naemia or **b**one lytic lesions. Other diagnostic criteria have been recently introduced, including at least 60% clonal plasma cells in the bone marrow, one or more focal bone marrow lesions on magnetic resonance imaging (MRI) imaging (indicating bone marrow involvement) or an involved to uninvolved free immunoglobulin light chain ratio > 100 [9].

The median survival of patients with MM has significantly improved since the 1990s, predominantly due to the introduction of new therapeutic agents [11,12]. These novel therapies include proteasome inhibitors (e.g., bortezomib and carfilzomib), immunomodulatory drugs (e.g., lenalidomide and pomalidomide) and monoclonal antibodies (e.g., daratumumab and elotuzumab, which target the CD38 and SLAMF7 proteins, respectively) [11,13]. In practice, these drugs are not used alone but rather in combination; however, despite significant improvements in quality of life and overall survival in the last few decades, relapsed disease inevitably occurs.

## 3. Lymphomas

Lymphomas are a diverse group of haemopoietic malignancies that arise from the clonal proliferation of lymphocytes, usually in lymph nodes. Characterisation of lymphomas morphologically, immunophenotypically and genetically allows for the identification of many different subtypes with varying prognostic and treatment algorithms, however they are generally classified as either Hodgkin lymphomas (HL) or non-Hodgkin lymphoma (NHL) [14]. HLs can be identified in lymph nodes by the presence of Reed–Sternberg cells admixed with variable numbers of B cells, T cells and other haemopoietic lineages. These lymphomas tend to be highly chemo-sensitive, with a 5-year survival rate exceeding 80% [15]. NHLs constitute a large group of diverse lymphoid tumours of either B or T-cell origin, with B-cell NHLs being more common than T-cell NHLs.

As an example, diffuse large B-cell lymphoma (DLBCL) is the most common non-Hodgkin lymphoma (NHL), accounting for 30–40% of all NHL cases and approximately one-third of all newly diagnosed lymphoma cases worldwide [16,17]. A treatment regimen consisting of cyclophosphamide, doxorubicin, vincristine and prednisone (CHOP) combined with the monoclonal antibody rituximab (R-CHOP) is the current frontline treatment for DLBCL [16]. While over 60% of patients can be cured with R-CHOP, those with either primary refractory disease or who relapse after a period of remission have poorer outcomes [16]. DLBCL can be further classified into two main molecular subtypes, activated B-cell (ABC) and germinal centre B-cell (GCB), the former being associated with significantly worse prognosis and complete remission (CR) rates of 30% and 70%, respectively [17]. In approximately 10% of cases, translocations involving *MYC* and *Bcl-2* and/or *Bcl-6* (double and triple hit DLBCL) are identified, which are also associated with less favourable clinical outcomes [18,19]. Furthermore, overexpression of *Bcl-2* and *MYC* (‘double-expressor’) can occur through mechanisms that do not involve chromosomal translocations [18,20]. It is for these patients with poorer clinical outcomes that novel therapeutics are urgently required.

## 4. Leukaemias

Leukaemias are a heterogenous group of diseases that arise from the clonal proliferation of immature or mature leukocytes [21]. It is estimated that 5202 people will be diagnosed with leukaemia in 2022 in Australia, with a 5-year survival rate of 64.4% [10]. Leukaemias can be characterised by the cell of origin (lymphoid or myeloid) and by the rate of proliferation (acute or chronic) [21]. The predominant subtypes of leukaemia and their prevalence in Australia are outlined in Table 1.

Chemotherapy, chemoimmunotherapy and stem cell transplantation are common treatment options for leukaemia, but with a better understanding of the mechanisms that drive these diseases, novel targeted agents are becoming more widely used. Response and outcome rates vary significantly between the different forms of leukaemia, but invariably a proportion of patients will relapse with refractory disease. Whilst survival rates among patients diagnosed with acute leukaemia have improved markedly in the last few decades, particularly for patients under the age of 50 [22], those diagnosed with acute lymphoblastic leukaemia (ALL) or acute myeloid leukaemia (AML) experience decreased survival with age with 5-year survival rates being as low as 12% and 4%, respectively, for people over the age of 80 at diagnosis [22].

Unlike patients with acute leukaemia, who generally require intensive treatment, patients with chronic leukaemia can often be managed with a ‘watch and wait’ approach with minimal to no treatment required for many years [21]. Approximately 30% of patients diagnosed with chronic lymphocytic leukaemia (CLL) never require treatment, whereas many CLL patients will have or will rapidly develop the symptomatic disease after diagnosis, requiring early intervention [23]. Chronic myeloid leukaemia (CML) is a slow-growing malignancy characterised by the Philadelphia chromosome formed by a reciprocal translocation of chromosomes 22 and 9 [24]. The resulting fusion oncogene (BCR-ABL) is effectively targeted by tyrosine kinase inhibitors such as imatinib, which have significantly improved survival rates among CML patients [24].

## 5. Ferroptosis

Regulated cell death is a fundamental physiological process that ensures cell integrity and homeostasis. Apoptosis is arguably the most well-studied form of cell death, but relatively recently, a study by Dixon et al. described an iron-dependent form of cell death associated with an accumulation of lipid peroxides, which they termed ferroptosis [1]. Ferroptosis is distinct morphologically, genetically and biochemically from other forms of cell death, including apoptosis, as it can be inhibited by iron chelation and lipophilic antioxidants [25]. Cells undergoing ferroptosis have a characteristic “ballooning phenotype” with an enlarged, empty cytoplasm, in contrast to the small, shrunken appearance of apoptotic cells with distinct membrane blebbing that precedes apoptotic body formation [26,27]. Ferroptotic cells also lack key morphological changes associated with other forms of programmed cell death, such as apoptotic bodies or the autophagosomes associated with apoptosis and autophagy, respectively [26]. Other features of ferroptosis, including reduced mitochondrial volume, increased mitochondrial membrane density and the absence of mitochondrial cristae, can also be observed using transmission microscopy [1]. 

Since its discovery, ferroptosis has been associated with many biological processes involved in cellular homeostasis, iron and amino acid regulation and the metabolism of polyunsaturated fatty acids [2]. In the context of cancer, induction of ferroptosis has the potential to be a novel treatment strategy, particularly for patients who relapse with drug-resistant disease following treatment with standard therapies. As cancer cells generally have higher levels of reactive oxygen species (ROS) due to increased metabolic and growth rates, cell death mechanisms such as ferroptosis that further elevate ROS levels may be a particularly effective and specific approach for the treatment of a range of cancers [28]. In addition to the aforementioned morphological changes, ferroptosis can be distinguished from other forms of cell death by monitoring the accumulation of lipid peroxides, using fluorescently labelled fatty acid analogues (i.e., C11-BODIPY), and through inhibition of cell death by either iron chelation (i.e., deferoxamine, DFO) or lipophilic antioxidants (liproxstatin-1, ferrostatin-1 or vitamin E) [29]. Furthermore, ferroptosis cannot be prevented by inhibitors of apoptosis, necroptosis or autophagy (i.e., z-VAD-FMK, necrostatin-1s and chloroquine, respectively) [1].

### 5.1. The Role of Iron in Ferroptosis

Iron is essential for cellular homeostasis [30], with key roles in oxygen transport, oxidative phosphorylation and DNA biosynthesis [31]. As iron chelation inhibits ferroptosis, this form of cell death is also clearly an iron-dependent process [1]. Intracellular iron levels are primarily regulated by the iron-storage protein ferritin and the transferrin receptor (TfR), which shuttles transferrin-bound iron into the cell through receptor-mediated endocytosis. The level of non-protein bound iron (labile iron pool) has implications in ferroptosis as labile iron reacts with hydrogen peroxide inside cells, yielding highly reactive hydroxyl radicals in a process known as the Fenton reaction [32]. These radicals indiscriminately damage all surrounding organic material within a range of a few nanometres, resulting in cellular damage (Figure 1) [32]. Iron also plays a role in ferroptosis through its actions on a group of iron-containing enzymes that mediate lipid peroxidation, known as lipoxygenases (LOXs) [2,33]. The key role of these enzymes is demonstrated by the LOX inhibitor, zileuton, which confers resistance to ferroptotic cell death in HT22 neuronal cells [34]. Furthermore, genetic knockdown or pharmacological inhibition of arachidonate lipoxygenases (ALOXs) protects cells against ferroptosis induced by erastin [3].

### 5.2. Lipid Peroxidation

Lipid peroxidation is a characteristic feature of ferroptosis, distinguishing it from other forms of programmed cell death. Phospholipid peroxidation results in the oxidative degradation of lipids, the formation of peroxyl radicals and damage to the plasma membrane (Figure 1) [35,36]. Sensitivity to ferroptosis is associated with the degree of lipid saturation, the cellular location of the phospholipids and the number of phospholipids containing polyunsaturated fatty acids (PUFA) relative to those containing monounsaturated fatty acids (MUFA) [2]. Phospholipids containing PUFAs are more readily oxidised, and therefore, supplementation with PUFAs such as arachidonic acid (AA) and linoleic acid (LA) increases the sensitivity of cancer cells to ferroptosis [3,37,38]. Other studies support this notion that cellular sensitivity to ferroptosis is associated with their lipidomic profile, demonstrating that the incorporation of MUFAs into phospholipids reduces the generation of lipid ROS in membranes and therefore protects against ferroptosis [3,39]. MUFAs can also displace PUFAs from intracellular phospholipids, including in the plasma membrane, reducing the potential for lipid ROS accumulation [39].

### 5.3. System x_c_− and GPX4

xCT, encoded by the *SLC7A11* gene, is a major part of the system x_c_- cystine/glutamate antiporter. The regulatory component, SLC3A2, is involved in many other solute transporter systems, and so, xCT has been the focus of much of the work into system x_c_-. Glutamate is pumped out of the cell in exchange for cystine, which is rapidly reduced to cysteine [40]. Cysteine is a rate-limiting factor in the production of glutathione (GSH), an important antioxidant due to its role as a cofactor of glutathione peroxidase 4 (GPX4) [41]. GPX4 is the primary enzyme that reduces phospholipid hydroperoxides into their corresponding alcohols, inhibiting lipid peroxidation and subsequent ferroptosis [2,42]. When the activity of GPX4 is inhibited, either directly or through GSH depletion, and lipid peroxidation exceeds tolerable levels, ferroptotic cell death is initiated (Figure 1) [2,43]. 

Class 1 ferroptosis inducers deplete cellular GSH by inhibiting xCT, therefore preventing cystine uptake (e.g., erastin) or interfering with GSH synthesis (e.g., buthionine sulfoximine, BSO). In vivo studies of erastin have been limited by its pharmacokinetics and poor solubility, but its more soluble analogue, imidazole ketone erastin (IKE), is metabolically stable and 100× more effective than erastin in some cell lines [44,45]. (1S,3R)-RSL3 (hereafter referred to as RSL3) was first described as a ferroptosis inducer in 2008 and shown to induce a similar phenotype to erastin via GSH-independent mechanisms [46]. In addition to RSL3, another frequently described class 2 ferroptosis inducer is ML162 [47,48]. Both RSL3 and ML162 are covalent inhibitors of GPX4 that bind and inhibit the protein directly [49].

### 5.4. Ferroptosis Suppressor Protein 1 and the Mevalonate Pathway

Ferroptosis has also been observed independent of GPX4 inhibition, likely due to the presence of other intracellular antioxidant systems. Recently, apoptosis-inducting factor mitochondrial 2 (AIFM2), which has since been renamed ferroptosis suppressor protein 1 (FSP1), has been implicated in resistance to ferroptosis [50,51]. A study by Doll et al. demonstrated that knockout of *FSP1* increased the sensitivity of melanoma (MDA-MD-435S), colorectal cancer (SW620), glioblastoma (U-373), lung cancer (A549 and NCI-H1437) and breast cancer (MDA-MB-436) cells to RSL3 [51]. Furthermore, resistance to RSL3 could be restored by re-expression of mouse *FSP1* in MDA-MB-436 breast cancer cells [51]. *FSP1* knockout in an osteosarcoma cell line (U-2 OS) did not affect GSH levels, suggesting its mechanism of action is independent of xCT or GSH synthesis [50]. The mechanism by which FSP1 mediates resistance to ferroptosis is thought to involve coenzyme Q_10_ (CoQ, also known as ubiquinone), generated by the mevalonate pathway. The mevalonate pathway is a multifaceted biological pathway that leads to the production of isopentenyl pyrophosphate (IPP) as well as CoQ (Figure 2). IPP is involved in the maturation of selenocysteine, an amino acid required for GPX4 translation [25,52]. IPP can also be converted to farnesyl pyrophosphate, which is an important upstream substrate in the generation of CoQ [53]. CoQ is a naturally occurring quinone that is vital to cell and tissue health in most aerobic organisms [54]. Studies have shown that FSP1 reduces CoQ to CoQ_10_-H_2_ (ubiquinol), which is a radical-trapping antioxidant that prevents the accumulation of lipid peroxides associated with ferroptosis (Figure 2) [50,51]. 

## 6. Clinical Applications of Ferroptosis

### 6.1. Ferroptosis in MM

Studies have already shown that treating MM cells with high concentrations of iron induces cell death, and this can be rescued by the ferroptosis inhibitor ferrostatin-1 [55]. Despite this, MM cells were shown to be more resistant to erastin-induced ferroptosis compared to another B-cell malignancy, DLBCL, and further investigation into this variation in sensitivity is required [43]. A recent study demonstrated that levels of GPX4 and xCT are higher in MM plasma cells than their healthy counterparts, suggesting MM plasma cells may be utilising this antioxidant pathway to withstand elevations in ROS levels [56]. Increased expression of these two proteins and their crucial roles in ferroptosis suggests they may represent drug targets for the treatment of MM. Furthermore, high expression of ferroptosis suppressor genes was correlated with reduced progression-free and overall survival in MM patients [57]. A number of groups have recently developed ferroptosis-related gene signatures that can be used to predict MM patient prognosis [58], high-risk patients [59] and/or drug sensitivity [60]. Fu et al. went on to show that erastin and doxorubicin synergistically reduced NCI-H929 and RPMI-8226 MM cell viability via GPX4 degradation and subsequent ROS generation [58]. Similarly, ferroptosis induced by GPX4 inhibition (RSL3 or ML162) synergistically decreased proliferation when combined with bortezomib or lenalidomide in RPMI-8226 MM cells [60]. Taken together, it can be seen that genes involved in ferroptosis may be a useful tool to prognosticate MM patients, while ferroptosis-inducing drugs have promise as agents to enhance existing chemotherapies used in the clinic.

The synthetic amino acid BSO has been found to reduce GSH levels in MM cells by inhibiting glutamate–cysteine ligase (GCL), the first enzyme in the GSH synthesis pathway, and increase the efficacy of bortezomib [61]. In bladder carcinoma cells, expression of xCT was found to be upregulated following bortezomib treatment in an Nrf2- and ATF-dependent manner [62]. Since the expression of both Nrf2 and ATF4 transcription factors is associated with poor prognosis and drug resistance in patients with bladder cancers, inhibition of xCT may represent an effective treatment to increase the efficacy of proteasome inhibitors [62]. Further supporting this, GSH degradation by omega-3 fatty acids increased the effects of bortezomib in OPM2 MM cells, and transcriptomic pathway analyses of the treated cells revealed changes in several gene pathways, including ferroptosis [63]. High-dose iron (600 µM ferric ammonium citrate) has also been shown to induce MM cell death and increase the efficacy of bortezomib in vitro and the bortezomib, melphalan and prednisone regimen in a MM mouse model [55]. These effects were again shown to involve ferroptosis in vitro, with increased production of the lipid oxidation by-product MDA and inhibition of cell death by ferrostatin-1 [55]. 

Lipidomic analyses have shown that AA levels are decreased in the plasma cells of patients with preclinical and early-stage MM [64]. This was confirmed in another analysis showing that although MM patients had higher levels of n-6 PUFAs compared to healthy controls, overall AA levels were reduced [65]. An earlier study, which utilised fluorescently-tagged AA, demonstrated that this fatty acid is readily taken up by MM cells and is primarily incorporated into triglycerides and phospholipids but that this uptake had no effect on proliferation [66]. The mode of transport has since been described to involve fatty acid transporters (FATPs) [67]. More recently, the addition of exogenous AA was associated with dose- and caspase-dependent apoptotic cell death in three MM lines but not healthy human peripheral blood mononuclear cells (PBMCs) [68]. Culturing cells in the presence of AA has also been shown to induce ferroptosis, which could be reversed by ferrostatin-1; the addition of physiological concentrations of AA was shown to decrease the proliferation and viability of primary MM cells and cell lines, with a concomitant decrease in GPX4 expression [69]. It is important to note that while high concentrations of AA can induce death in MM cells, low concentrations have been shown to promote their proliferation. In fact, MM cells induce lipolysis in bone marrow adipocytes and upregulate FATPs in the presence of free FAs [67]. This highlights the need to better understand the physiological concentrations of AA that are required to induce cell death and improve delivery systems to ensure levels do not drop below the threshold such that cancer cell proliferation is promoted.

With the advent of large-scale, high-throughput drug screening technologies, there has been a rapid increase in the number of drugs that are now known to induce ferroptosis. FTY720 was first developed by structurally modifying the antibiotic myriocin and is now approved by the FDA for the treatment of multiple sclerosis [56]. Initial studies of FTY720 in MM suggested that this compound induces apoptosis and autophagy [70]; however, a more recent study showed that cell death in MM cells following treatment with FTY720 is associated with an accumulation of ROS and can, at least partially, be inhibited by ferrostatin-1 [56]. Furthermore, the study showed that FTY720 reduced expression of GPX4 and xCT in vitro, both at the mRNA and protein levels and concluded that the drug likely induces ferroptosis and autophagy, mediated by the protein phosphatase 2A/AMP-activated protein kinase pathway [56]. The naturally occurring flavone, apigenin and extracts from the plants *Thymus vulgaris*, *Arctium lappa*, *Fumaria officinalis* L. and *Lithospermum erythrorhizon* were also shown to induce cell death in a range of MM cell lines [71,72,73,74]. Cell death could be at least partially inhibited with ferrostatin-1, liproxstatin-1 or deferoxamine in all cases, suggesting ferroptosis was involved in the observed response and indicating that naturally derived plant extracts may be useful tools to induce ferroptosis in MM cells.

Recent research in MM has also uncovered a link between ferroptosis and subsequent DNA changes [75]. Induction of ferroptosis in MM results in the upregulation of a multitude of key genes involved in cellular stress, cell death, inflammation and fatty acid metabolism, including the ferritin heavy chain 1 (*FTH1*), ferritin light chain (*FTL*), *HO-1* and *SLC7A11* genes [75]. Of 616 differentially expressed genes identified in MM cells undergoing ferroptosis, the upregulation of 95 genes was inhibited by pre-treating cells with ferrostatin-1 [75]. The 95 genes identified included those that encode zinc finger proteins and genes with roles in metal binding, nuclear receptor signalling and chromatin remodelling [75]. 

Taken together, these findings demonstrate that MM cells have the capacity to undergo ferroptosis; however, further studies are required to identify combinations and concentrations of ferroptosis-inducing agents that are achievable in vivo to harness the potential of ferroptosis as a clinical strategy.

### 6.2. Ferroptosis in Lymphoma

DLBCL cells have been shown to be significantly more sensitive to erastin-induced growth inhibition compared to MM and AML cells [43]. This may be explained by the dependence of DLBCL cells on xCT for cysteine uptake, as they are unable to synthesise cysteine from methionine via the transsulfuration pathway [44]. Furthermore, a number of ferroptosis gene signatures, including genes such as *FTH1*, *GPX4*, *STEAP3*, *LPCAT3* and others, have been developed to predict prognosis and overall survival in patients with DLBCL [76,77,78]. Specifically, the expression of GPX4 was shown to be an independent marker of poor prognosis in DLBCL [79]. Immunohistochemistry was used to examine GPX4 expression in samples from 93 DLBCL patients, and the GPX4 positive group (35%) had significantly lower overall and progression-free survival rates [79]. Interestingly, no significant changes in GPX4 mRNA were observed, suggesting that GPX4 protein expression is regulated by post-transcriptional modification [79]. These findings support the idea that GPX4 may confer a survival advantage on DLBCL cells, possibly through oxidant tolerance and decreased sensitivity to ferroptosis, and that inhibition of GPX4 may represent a potential therapeutic target for patients with high-risk DLBCL disease. A recent study found that treatment of DLBCL cells with dimethyl fumarate depletes the cells of GSH and retards their proliferation and that these effects were potentiated by inhibition of FSP1 [80]. However, significantly higher levels of lipid peroxidation were observed in DLBCL cells classified as GBC than ABC following treatment with dimethyl fumarate, which may be due to elevated levels of 5-lipoxygenase in the GCB sub-type [80]. 

Propolis, a resinous product from Western honeybees, has been shown to have anti-tumour properties in a range of cancer types. Recently, extracts from Chinese propolis were shown to induce cell death in the SU-DHL-2 DLBCL cell line, and proteomic changes following drug incubation revealed ferroptosis as the most differentially expressed pathway [81]. The propolis ethanol extract was analysed by ultra-performance liquid chromatography–electrospray ionization mass spectrometry (UPLC-ESI-MS) and determined to contain a variety of flavonoids, phenolic compounds, acylated glycerol and fatty acids. Included in the list of compounds was apigenin which has previously been shown to induce ferroptosis on its own or as a compound in other plant extracts in MM and leukaemic cell lines [71,72,73]. Another natural compound kayadiol, which is extracted from the pulp of a Japanese tree (*Torreya nucifera*), induced death in extranodal natural killer/T-cell lymphoma (NKTCL) cell lines and peripheral blood lymphocytes (PBLs) extracted from NKTCL patients but not healthy donor PBLs [82]. 

Sensitivity to ferroptosis varies between different DLBCL cell lines, with the IC_50_ of the erastin analogue, IKE, ranging from 1 nM to almost 100 µM [44]. Consistent with ferroptosis, cell death induced by IKE was associated with a significant increase in levels of lipid peroxidation and MDA and could be inhibited by pre-treating cells with ferrostatin-1 [44]. Similar effects were also observed in a mouse lymphoma model and were associated with a decrease in GSH levels from as early as 4 h after IKE dosing [44]. Decreased tumour volume in mice treated with IKE was consistent with increased MDA levels, which is indicative of lipid peroxidation and suggestive of ferroptosis as a mechanism of cell death. As single agents, the PUFAs AA, eicosapentaenoic acid (EPA) and γ-linolenic acid all caused a decrease in cell viability in the ferroptosis-sensitive DLBCL cell line SU-DHL-6 at high concentrations approaching 100 µM, whereas a lower concentration of 10 µM was sufficient to sensitise the cells to IKE [44]. Conversely, the addition of the MUFAs, oleic acid or palmitoleic acid protected the cells from ferroptosis, which is consistent with earlier reports demonstrating that addition of oleic acid to ferroptosis-sensitive BJeLR (human skin tissue), HT-1080 (fibrosarcoma) and G-401 (rhabdoid kidney tumour) cell lines can prevent RSL3-induced ferroptosis [3,44]. 

As discussed earlier, iron homeostasis plays a critical role in ferroptosis, and therefore targeting pathways that regulate iron levels may represent an effective treatment approach. A recent study in DLBCL with the small lysosomal iron-targeting molecule, ironomycin, demonstrated that the drug sequesters iron within lysosomes, which leads to the generation of lysosomal ROS, dysfunction of lysosomes and cell death [83]. The observed mechanism of cell death was suspected to be ferroptosis, as levels of cellular GSH were depleted, and the cell death was partially prevented by ferrostatin-1 but not by the necroptosis or apoptosis inhibitors, necrostatin-1 or z-VAD-FMK [83,84], respectively.

Recent studies demonstrate that the cytotoxicity of APR-246 (eprenetapopt), a prodrug that binds mutant *TP53*, is iron-dependent and can be prevented by iron chelators or inhibitors of lipid peroxidation, but not necroptosis, pyroptosis or apoptosis inhibitors in DLBCL cells [85]. Interestingly, the autophagy inhibitor chloroquine was able to prevent APR-246-induced death in OCI-Ly7 cells with mutant *TP53* (missense in exon 7) but not any other DLBCL cell lines. The binding of APR-246 to mutant *TP53* restores the ability of the transcription factor to interact with target genes and, as a result, induces ferritinophagy. APR-246 is also effective at inducing ferroptosis in cells with WT *TP53* through mechanisms thought to be independent of *TP53* [85]. In *Eµ-Myc* mouse lymphoma cells, APR-246 induces apoptosis regardless of *TP53* status but was, however, more effective in WT cells. Ferroptosis was not observed in these cells following APR-246 treatment, but Fer-1 could partially prevent cell death in histiocytic lymphoma cells (BAX/BAK/MLKL KO U937 cells) [86]. Other cancer cell lines and cell death mechanisms were investigated in this study leading to the conclusion that APR-246 can induce a variety of cell death types, including ferroptosis, regardless of *TP53* status [86].

Using computational and experimental approaches, altretamine, an FDA-approved anti-cancer drug used to treat ovarian cancer, was shown to have a similar mechanism of action as the drug sulfasalazine [87]. Unlike sulfasalazine, however, altretamine did not deplete GSH levels in the U2932 DLBCL cell line but did induce a significant increase in oxidised phospholipids, implicating GPX4 as a target of this drug [87]. Artesunate, a derivate of the plant extract artemisinin, is approved by the FDA for malaria treatment and its anticancer effects have been explored and recently revealed to involve ferroptosis. In Burkitt’s lymphoma cells, only inhibitors of ferroptosis but not necroptosis or apoptosis could prevent artesunate-induced cytotoxicity. Furthermore, artesunate activity was associated with an increase in general ROS and lipid peroxidation and was revealed to involve the ATF4-CHOP-CHAC1 pathway [88]. Similarly, adult T-cell leukaemia/lymphoma (ATLL) cells in vitro and in a mouse xenograft model underwent ferroptosis as well as caspase-dependent apoptosis and cell cycle arrest following treatment with artesunate [89]. In a panel of DLBCL cell lines and a xenograft mouse model, artesunate induced apoptosis, autophagy and ferroptosis through inhibiting STAT3 signalling, an important pro-survival pathway in cancer cells [90]. Further investigation of already FDA-approved drugs, such as altretamine and artesunate, is warranted, given their involvement in ferroptosis and already known safety and pharmacokinetic profiles.

### 6.3. Ferroptosis in Leukaemia

#### 6.3.1. Acute Lymphoblastic Leukaemia

ALL cells were shown to undergo ferroptosis following treatment with RSL3 or erastin in combination with the Smac mimetic BV6, which binds proteins that inhibit apoptosis [91]. It has also been shown that lipoxygenases play an important role in ferroptosis in ALL cells, as the lipoxygenase inhibitors, Baicalein, NDGA, Zileuton or PD146176, prevent RSL3-induced cell death [92]. The sensitivity of ALL cells to ferroptosis may be explained by hypermethylation of *FSP1* and downregulation of FSP1 expression at both the mRNA and protein level, which has been shown in several ALL cell lines and patient samples [93]. Furthermore, elevated *FSP1* expression was correlated with significantly worse overall survival in AML but has not yet been demonstrated in ALL [93]. Recent studies have shown that ferroptosis may also be involved in the efficacy of a number of natural plant extracts against ALL cells that are resistant to standard therapeutic agents and difficult-to-treat T-ALL cells [94,95,96,97].

#### 6.3.2. Acute Myeloid Leukaemia

There are extensive studies focusing on the potential role of ferroptosis in the treatment of AML. These include investigating the effects of combining erastin with chemotherapy drugs [98] or the ferroptotic effects of other agents, such as the antimalarial drug dihydroartemisinin [99,100], a novel all-trans retinoic acid derivative [101] and the circRNAs circKDM4C, which upregulates ferroptosis [102] and circZBTB46 which prevents ferroptosis [103]. Natural derivatives with ferroptosis-inducing potential in AML have also been identified including a pollen flavonoid extract, typhaneoside [104], a monoterpenoid plant extract, perillaldehyde [105], or the diterpenoid epoxide plant extract, triptolide, which induces ferroptosis to sensitise AML and CML cells to doxorubicin [97]. Inhibitors of aldehyde dehydrogenase 3a2 and GPX4 have also been shown to synergise and induce cell death via ferroptosis in AML cells, both in vitro and in vivo [106]. 

As discussed earlier, APR-246 induces ferroptosis in DLBCL cells, and this has also been demonstrated in AML cells, independent of *TP53* mutational status [85,107]. Through several methods, using various cell death inhibitors and measuring markers of necroptosis (MLKL phosphorylation) and apoptosis (caspase cleavage), it was determined that APR-246 only induced ferroptosis in AML cells [107]. Furthermore, APR-246 synergised with other ferroptosis inducers in vitro, and while it was ineffective as a single agent in vivo, it effectively reduced tumour burden in mice engrafted with *SLC7A11* KD AML cells [107]. A number of phase II and III clinical trials have investigated APR-246 in combination with azacitidine for patients with AML or myelodysplastic syndromes (MDS) and mutated p53 (Phase Ib/II Clinical Trials Identifiers: NCT03072043 and NCT03588078, Phase III Clinical Trials identifier: NCT03745716). While ferroptosis was not specifically investigated in these clinical trials, the drugs were well tolerated, and APR-246 enhanced the anti-tumour effects of azacytidine [108,109,110].

A study by Akiyama et al. demonstrated that high expression of the *SLC7A11* or *GPX4* genes is associated with significantly shorter overall survival rates in AML patients [111]. Zhong et al. also found that along with other ferroptosis-related genes found to be differentially expressed in AML patients, *GPX4* overexpression was again associated with poor prognosis in other publicly available cohorts [112]. This overexpression of GPX4 in AML patients highlighted a potentially efficacious drug target in this cancer, and both studies found that inhibition of GPX4 with ML210 or RSL3 induced the death of AML cells in vitro [111,112]. This is supported by other studies, which have also shown that higher expression of *SLC7A11*, *GPX4* and *LPCAT3* is associated with a worse prognosis in AML, while overexpression of *TFRC*, which encodes the transferrin receptor, was found to protect cells against ferroptosis [113,114]. In children with non-M3 AML, *FTH1* expression is associated with a poor prognosis and is upregulated during the proliferation of AML cell lines [115]. Numerous publications have identified other ferroptosis-related gene signatures, eluding to the relevance of ferroptosis in AML but also indicating the dataset used and other factors may contribute to what genes can be used to predict both prognosis and ferroptosis-sensitivity in AML [116,117,118,119,120,121].

#### 6.3.3. Chronic Lymphocytic Leukaemia 

Primary CLL cells express low levels of xCT and rely on cysteine produced by bone marrow stromal cells to reduce levels of intracellular ROS [122]. The cysteine taken up by CLL cells fuels the production of GSH, which protects the cells from the cytotoxic effects of fludarabine and oxaliplatin. Inhibitors of xCT or depletion of GSH significantly increased the sensitivity of CLL cells to both fludarabine and oxaliplatin [122]. Although the study by Zhang et al. was conducted prior to the description of ferroptosis, the effects of the xCT inhibitor and GSH depletion on cell viability strongly suggest that the cytotoxic effects observed involved ferroptosis. More recently, a study on primary CLL cells and cell lines also showed that treating cells with iron in combination with Bruton’s tyrosine kinase (Btk) inhibitor, ibrutinib or the BCL-2 inhibitor, venetoclax, led to a significant accumulation of MDA and death of leukemic, but not healthy, cells [123]. Although apoptosis is understood to be the predominant mechanism of action of both ibrutinib and venetoclax, ferroptosis may also play an important role and potentiate the efficacy of these drugs. A study of a small cohort of 36 patients with CLL identified a ferroptosis-related prognostic score whereby nine genes were associated with worse overall and treatment-free survival [124]. Further mechanistic studies are required to understand whether these genes have a ferroptosis-related function in CLL patients on top of their prognostic value.

#### 6.3.4. Chronic Myeloid Leukaemia 

Cysteine depletion in an imatinib-resistant CML cell line (K562/G01) but not WT parental cells (K562) induces ferroptosis [125]. This ferroptosis was associated with the upregulation of the gene encoding thioredoxin reductase 1 (TXNRD1) in K562/G01 cells, a member of the thioredoxin (Trx) system, an important antioxidant and redox regulator [125]. Despite an increase in gene expression, the activity of TXNRD1 was decreased, which was thought to be a result of negative feedback regulation [125]. These findings led the researchers to investigate a shRNA-mediated knockdown of TXNRD1 in WT K562 cells. In these knockdown cells, cysteine depletion caused a decrease in viability which could be prevented by Fer-1, as well as increased ROS and MDA and morphological changes consistent with ferroptotic cell death [125]. The effect of TXNRD1 KD on increasing sensitivity to cysteine depletion suggests the Trx system may play a role in regulating ferroptosis. The drug tetrandrine citrate kills CML cells in vitro and reduces tumour growth in an imatinib-resistant mouse xenograft model [126], and while ferroptosis was not confirmed as the mechanism of cell death, subsequent studies have shown that tetrandrine citrate does induce ferroptosis in breast cancer cells [127]. 

## 7. Potential Nanotechnologies for Induction of Ferroptosis

Emerging nanotechnologies have the potential to significantly improve the targeting, delivery and pharmacokinetic behaviour of drugs while reducing toxicities [128]. Drug delivery is an important determinant of efficacy in the treatment of cancer and has been a limiting factor in the development of therapeutic options, including those that target the ferroptosis pathway. Large molecules pose many issues in drug delivery, including poor solubility, poor absorption, off-target binding and in vivo instability. In the context of ferroptosis, primarily only in vitro studies of erastin and RSL3 have been possible due to poor solubility and toxicities in vivo [41,129]. Although cancer cells are generally more sensitive to ferroptosis, effects against healthy tissue have also been significant. To harness ferroptosis as an approach for cancer therapy, we must develop new means of specifically targeting the tumour cells [130]. Ferroptosis-inducing nanotechnologies have been investigated in cancers of the breast, colon and lung, as well as for neuroblastoma and hepatocellular carcinoma [131,132,133,134,135,136,137,138,139]. These studies primarily include the use of nanotechnologies such as dendritic mesoporous silica nanoparticles, iron oxide nanoparticles, micelles and liposomes (Figure 3). An important property to consider when designing nanotechnologies, particularly for the treatment of central nervous system (CNS) lymphoma, is their ability to cross the blood–brain barrier (BBB). This is a complex process and depends on nanoparticle size, shape and surface charge, amongst other factors [140]. Overall, in terms of size, nanoparticles up to approximately 20 nm are large enough to avoid renal excretion while being small enough to penetrate the BBB. Moreover, those nanoparticles with a negative (or relatively lower) zeta potential show greater transport through the BBB [140].

Dendrimers are a type of spherical synthetic polymer that has a structure comprised of repeated branching chains expanding from a central core that typically contains exterior functional groups (Figure 3) [143]. Therapeutic cargo can then be encapsulated within the inner space of the dendrimer or bound to the functional groups on the exterior of the polymer. These characteristics make dendrimers highly bioavailable and biodegradable, both of which are important qualities for an efficient drug delivery system [143]. The application of dendrimers in the context of ferroptosis was investigated in pancreatic carcinoma with promising results [144]. Generation 5 poly(amidoamine) (PAMAM) dendrimers were loaded with ferric iron by chelation to hydroxyquinoline-2-carboxylic acid (8-HQC) and plasmid DNA encoding p53. Gold nanoparticles were then entrapped within the inner space of the dendrimer, as this was previously shown to enhance gene delivery efficiency [144]. The dendrimers showed efficacy both in vitro and in vivo, effectively decreasing cell viability and proliferation. Furthermore, they were able to enhance p53 expression and trigger apoptosis while simultaneously inducing ferroptosis by inhibiting xCT and increasing ROS generation through the Fenton reaction [144].

Iron oxide nanoparticles (IONs) are iron-based structures most commonly synthesised through co-precipitation whereby ferric and ferrous iron are mixed at high temperatures at a molar ratio of 1:2 in highly basic solutions to obtain the nanoparticles Fe_3_O_4_ or γ-Fe_2_O_3_ [145]. IONs have traditionally been used as drug carriers and contrast agents for both clinical and pre-clinical purposes; however, they have also been used as iron supplements for patients with iron deficiency [146]. The simplistic nature of IONs yields many benefits, notably physical and chemical stability, biocompatibility and environmental safety [142]. The coating of IONs is also common practice, with outer shells, ranging from polymers to bioactive molecules, employed for functionalisation of the nanoparticle, improving stability, biodistribution and pharmacokinetics [142]. An example of an FDA-approved ION is ferumoxytol, used for iron replacement in anaemic patients, which has recently been investigated as an anti-tumour agent [146,147]. The involvement of ferroptosis in the mechanism of action of ferumoxytol is discussed further below.

Micelles, in their most basic form, are amphiphilic molecules (surfactants) arranged spherically in aqueous solutions [148]. Surfactants are classified according to the chemical nature of their polar head and typically contain long hydrocarbon, fluorocarbon or siloxane chains as their hydrophobic tail [148]. The versatility of micelles makes them fantastic for biomedical applications, particularly as drug delivery systems [148]. The two unmatched advantages of micelles when compared to other drug delivery systems are their size and the feasibility of large-scale manufacturing [149]. Gao et al. encapsulated RSL3 in micelles rich in unsaturated FAs, thereby supplementing cells with this known ferroptosis substrate while simultaneously delivering a ferroptosis-inducing agent [150]. The micelles were shown to be more effective at reducing tumour volume in doxorubicin-resistant human ovarian adenocarcinoma-bearing mice compared to drugs administered by more conventional means, and no adverse side effects were observed in mice treated with the micelles [150]. Another group demonstrated that electron-accepting RSL3-loaded micelles were able to reduce intracellular NADPH levels and induce ferroptosis effectively in vitro and in vivo [151]. The combination of electron-accepting micelles and RSL3 induced synergistic NADPH depletion and significantly decreased tumour mass in 4T1 tumour-bearing mice [151]. 

Liposomes are spherical nanoparticles that range in size from 30 nm to a few microns [143]. Liposomes consist of lipids that form a bilayer resembling that of the plasma membrane and represent a relatively safe and effective method of drug delivery [152,153]. The hydrophobic nature of the lipid bilayer in liposomal nanoparticles enables the incorporation of a wide variety of hydrophilic agents within the aqueous core (Figure 4). 

Liposomes are incredibly versatile as they can be modified with polymers, antibodies or proteins that determine specificity and uptake. For example, liposomes can be coated with antibodies against receptors that are known to be overexpressed on specific cancer cells, thus targeting drug delivery to a specific tumour site and sparing healthy tissue. In the context of ferroptosis, liposomes are particularly appealing as they can be manufactured from specific lipids and fatty acids that are involved in ferroptosis. Table 2 summarises the published literature on ferroptosis-inducing liposomes in solid tumours. Due to a paucity of ferroptosis-inducing, liposome-based nanotechnologies in haematological malignancies, we discuss ferroptosis-inducing nanotechnologies more broadly for these cancers below.

## 8. Ferroptosis Nanotechnologies in Haematological Malignancies

While there are a limited number of studies that examine the potential of nanotechnologies as a means of inducing ferroptosis in haematological malignancies, there are, however, numerous studies reporting the efficacy of non-ferroptotic nanotechnologies in these cancers, suggesting their utility in this area. One example is pegylated liposomal doxorubicin, or DOXIL, which was the first FDA-approved nanotherapeutic drug for cancer therapy [168]. DOXIL outperformed conventional doxorubicin in terms of clinical efficacy, pharmacokinetics, biodistribution, toxicity and overall improvement in patient quality of life [168]. While no studies to date have examined ferroptosis-targeting nanotechnologies in MM, other nanotechnologies, as well as DOXIL, have shown promise in this malignancy, for example, liposomes containing bortezomib [168,169,170,171].

Unlike MM, there are a number of papers investigating ferroptosis-inducing nanotherapies in lymphoma. Zhang et al. showed the effects of the erastin analogue, IKE, on the viability of DLBCL cell lines varied, with IC_50_s ranging from 1 nM to 100 µM, and that cell death and lipid oxidation following treatment with IKE could be prevented with ferrostatin-1 in SU-DHL-6 cells [44]. IKE was then encapsulated in polyethylene glycol-poly (lactic-co-glycolic acid) (PEG-PLGA) nanoparticles and used to treat mice bearing an SU-DHL-6 xenograft [44]. Although similar effects of free IKE and IKE-containing nanoparticles were observed on tumour size, weight loss which was used as a measure of toxicity, was only observed in mice treated with free IKE and not in mice treated with IKE nanoparticles. This toxicity was thought to be due to the precipitation of the drug in the peritoneal cavity, while IKE nanoparticles appeared to accumulate mainly in the tumour tissue rather than in the plasma [44]. 

Nanoparticles may also have the capacity to aid in the induction of ferroptosis by mechanisms other than through drug delivery. For example, a recent study demonstrated that high-density lipoprotein (HDL)-like nanoparticles resulted in cell death of cell lines and primary B-cells of Burkitt’s lymphoma, DLBCL and T cell–rich B cell lymphoma through mechanisms consistent with ferroptosis [172]. These HDL-like nanoparticles were manufactured to specifically target scavenger receptor type B1 (SCARDB1), a receptor that mediates cholesterol uptake and results in a compensatory increase in de novo cholesterol synthesis [172]. Consequently, increased cholesterol levels depleted the cells of GPX4 and initiated ferroptosis, which was confirmed using C11-BODIPY as a sensor of lipid ROS and with the ferroptosis inhibitor, ferrostatin-1 [172].

Ferumoxytol is an FDA-approved polyglucose sorbitol carboxymethyl ether-coated ION used for iron replacement which has recently been reported to have anti-tumour activity. Ferumoxytol inhibited cell proliferation in DLBCL cell lines while also inducing a dose-dependent reduction in cell viability, which was originally described as apoptosis. Upon further research, it was determined that ferumoxytol treatment induced an increase in phospholipid ROS by LiperFluo staining, suggesting that ferroptosis contributed to the observed cell death [146]. The mechanism of action of ferumoxytol involves the release of ferrous and ferric iron once the IONs are within macrophages in the liver, spleen and bone marrow, triggering the Fenton reaction and production of ROS [173]. Ferumoxytol treatment in DLBCL mice models inhibited the growth of tumours by inducing ferroptosis in a dose-independent manner, with no significant differences in mice body weight between treatments. Electron microscopy analysis of in vivo samples revealed mitochondrial membrane rupture and reduced mitochondrial cristae, suggestive of ferroptosis [146]. Similar to DLBCL, ferumoxytol induces oxidative stress and cell death in AML cells in vitro [147]. While ferumoxytol has only been approved by the FDA for the treatment of patients with chronic kidney disease and anaemia, the results of this study highlight the potential of ferroptosis-inducing IONs as a treatment for haematological malignancies.

Recently, nanoparticles loaded with a drug that targets N^6^-methyl-adenosine (m^6^a) RNA methylation were shown to be effective against AML cells both in vitro and in vivo [174]. These nanoparticles were modified to deplete AML cells of GSH and to target the leukaemic cells by conjugation to a peptide that recognises C-type lectin-like molecule-1 (CLL-1), which is overexpressed on AML blasts and stem cells. Cell death was associated with reduced GPX4 activity and increased levels of lipid peroxidation, suggesting ferroptosis was a key mechanism of action of these nanoparticles [174]. A similar approach was taken by Yu et al., who found that AML cells had higher GSH levels and GSH/GSSG ratio than normal haematopoietic cells in a mouse model [175]. They developed a GSH-responsive cysteine polymer-based ferroptosis-inducing nanomedicine (GCFN) and found that the nanomedicine caused GSH depletion through oxidation of GSH through the disulphide group in the cysteine polymer. An increase in MDA, BODIPY-C11 staining and the ability of Fer-1 to prevent death and cell proliferation inhibition indicated the involvement of ferroptosis. Furthermore, GCFN specifically targeted the bone marrow and spleen of an AML mouse model where leukemic cells are most abundantly found, while uptake in WT mice was mostly localised to the liver. In the bone marrow specifically, 97.6% leukaemic stem cells and 84.6% AML cells took up fluorescently labelled GCFN, while of the haematopoietic stem and progenitor cells (HSPCs) and other immune cell populations, less than 15% took up the nanomedicine. While this nanomedicine was not specifically generated to target leukaemic cells, it was thought that their proclivity for cysteine uptake was behind this cancer-targeting quality [175].

## 9. Conclusions

Haematological malignancies affect thousands of people worldwide every year, and despite many therapeutic advances, a significant proportion of patients will relapse with the drug-refractory disease. Ferroptosis represents a potential approach for treating a range of cancers, particularly those that do not respond to standard chemotherapies, which generally induce apoptosis. The ever-growing number of studies on ferroptosis since first being described in 2012 has greatly increased our understanding of this mechanism of cell death. However, further work is required to define how ferroptosis is regulated and determine why there is such variability in sensitivity between different cancers and sometimes even within the same cancer. This is crucial for the development of novel treatment approaches that harness the potential of ferroptosis.

The main factors limiting pre-clinical and clinical trials of ferroptosis-mediated therapies are the poor pharmacokinetics and toxicities associated with bona fide ferroptosis inducers. Nanotechnologies designed to target and precisely deliver drugs to tumour cells specifically may be one approach to improving specificity and increasing bioavailability. We have discussed many of the studies to date that have described different nanotechnologies that may be applicable in this context, including liposomes, which enable the targeted delivery of relevant lipids and encapsulated ferroptosis-inducing compounds to tumour cells. Combinations of ferroptosis-inducing compounds with current chemotherapies that are predominantly inducers of apoptosis have also been shown to have potent anti-tumour effects. This suggests that low doses of ferroptosis-inducing compounds may be effective when used in conjunction with existing chemotherapeutic regimens, possibly reducing toxicities and the development of drug resistance. It is envisaged that with further research, a class of ferroptosis-inducing, anti-cancer nanotherapeutics will find its place alongside other novel cancer drug classes, including monoclonal antibodies, antibody-drug conjugates and CAR-T cells. 

## Figures and Tables

**Figure 1 ijms-24-07661-f001:**
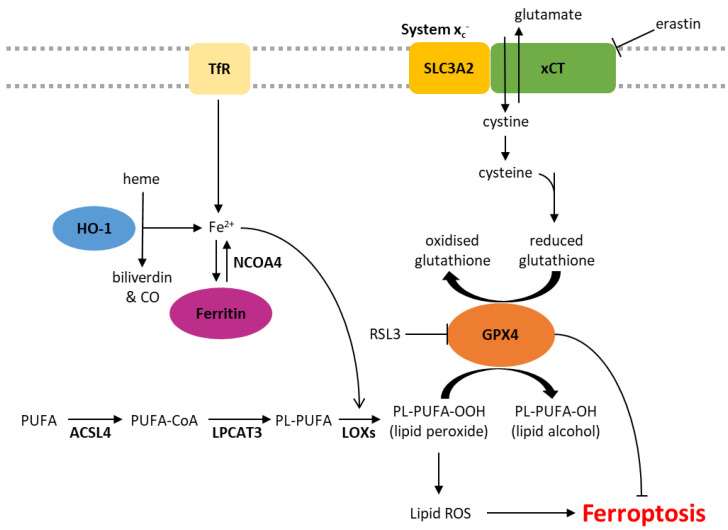
Biochemical pathways involved in the regulation of ferroptosis. ACSL4, acyl-CoA synthetase long-chain family member 4; CO, carbon monoxide; Fe^2+^, ferrous iron; GPX4, glutathione peroxidase 4; HO-1, heme oxygenase 1; LOXs, lipoxygenases; LPCAT3, lysophosphatidylcholine acyltransferase 3; NCOA4, nuclear receptor coactivator 4; PL, phospholipid; PUFA, polyunsaturated fatty acid; ROS, reactive oxygen species; TfR, transferrin receptor.

**Figure 2 ijms-24-07661-f002:**
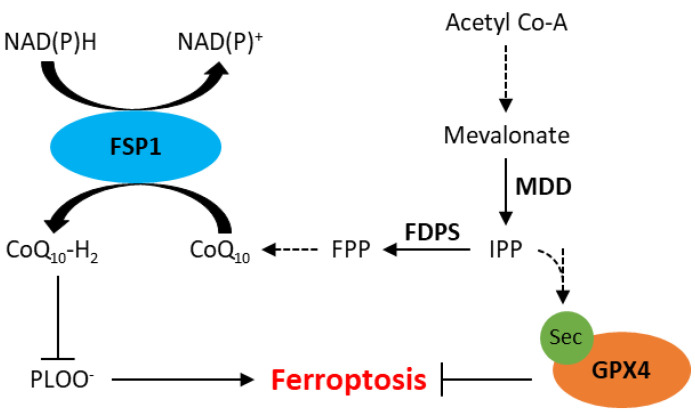
Schematic representation of the mevalonate pathway and its role in inhibition of ferroptosis. CoQ_10_, ubiquinone; CoQ_10_-H_2,_ ubiquinol; FDPS, farnesyl diphosphate synthase; FPP, farnesyl phosphate; FSP1, ferroptosis suppressor protein 1; GPX4, glutathione peroxidase 4; IPP, isopentenyl phosphate; MDD, mevalonate diphosphate decarboxylase; PLOO−, lipid peroxyl radicals; Sec, selenocysteine. Dotted arrows represent multiple steps within a pathway.

**Figure 3 ijms-24-07661-f003:**
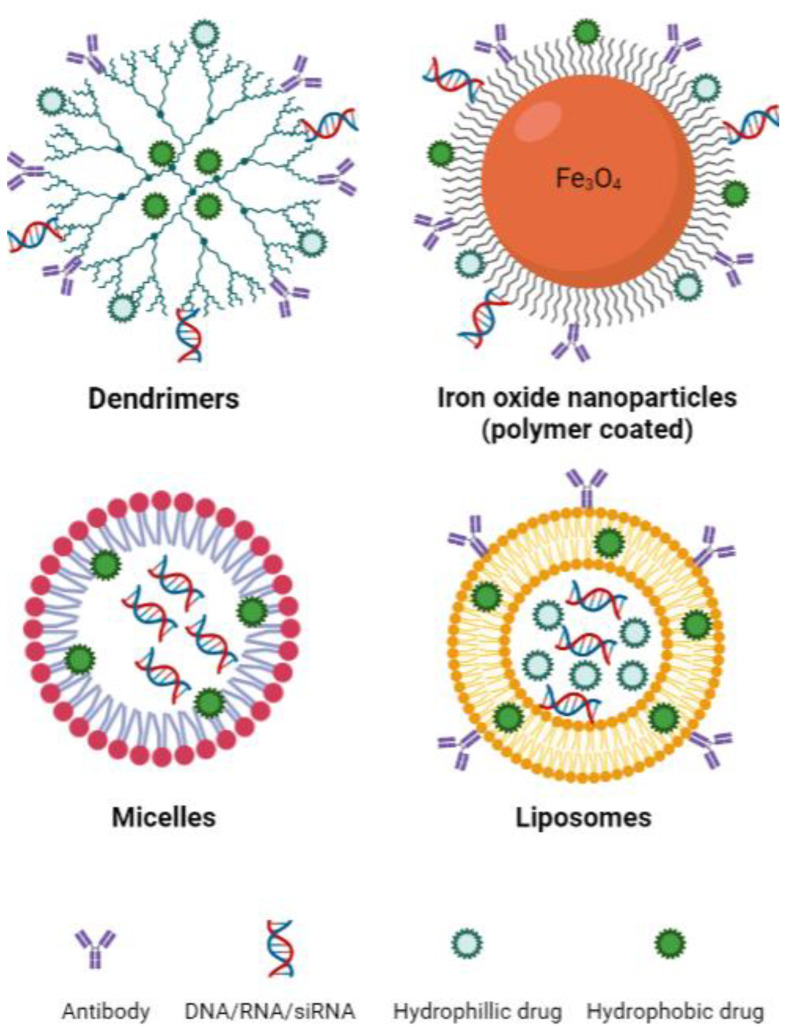
Basic structure of common nanotechnologies used to induce ferroptosis. Sizes: dendrimers 1–10 nm [141], iron oxide nanoparticles 10–20 nm [142], micelles 10–100 nm [143], liposomes 30 nm to several microns [143]. Created with BioRender.com (accessed on 22 March 2023).

**Figure 4 ijms-24-07661-f004:**
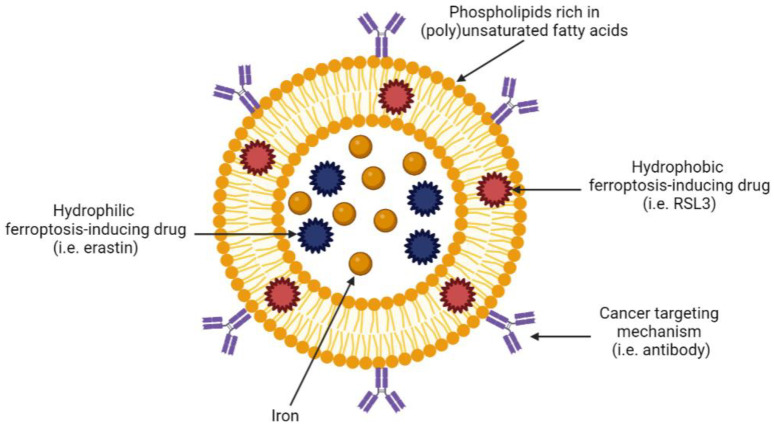
Example of a tumour-targeting, ferroptosis-inducing liposome. Created with BioRender.com (accessed on 27 March 2023).

**Table 1 ijms-24-07661-t001:** Leukaemia statistics in Australia.

Leukaemia Subtype	Median Age at Diagnosis *	Incidence *	Deaths ^+^	5-Year Survival Rate *
**Acute lymphoblastic leukaemia (ALL)**	15.1	382	90	74.5%
**Acute myeloid leukaemia (AML)**	69.1	1082	1169	26.4%
**Chronic lymphocytic leukaemia (CLL)**	71.5	1945	306	85.4%
**Chronic myeloid leukaemia (CML)**	61.4	392	84	83.4%

* Data extracted from 2018 Cancer data in Australia [10]. ^+^ Data extracted from 2020 Cancer data in Australia [10].

**Table 2 ijms-24-07661-t002:** Summary of recent liposome technologies aimed at inducing ferroptosis in non-haematological cancers.

Liposome Functionalisation/Contents	Malignancy	Findings	Ferroptosis Measure	Ref.
**Folate-modified LPs encapsulating erastin and MT1DP (lncRNA that represses antioxidation) (E/M@FA-LPs)**	Human NSCLC	E/M@FA-LPs more effective than erastin and MT1DP alone—decreased cell viability in vitro and reduced tumour volume/weight in vivo.	GSH depletion in vitro and increased MDA in vitro and in vivo.	[154]
**Unsaturated fatty acid-rich phosphatidylcholine LPs loaded with ferric ammonium citrate (LPOgener)**	Human breast cancer and murine mammary carcinoma	LPOgener effectively killed breast cancer cells, with some toxicity to normal liver cells, while FAC and empty LPs had no effect. A similar trend was seen in a mouse model, and no damage to any organs was observed.	Small mitochondria with condensed membranes in vitro. Increase in oxidised C11-BODIPY in vitro and in vivo.	[155]
**MMP2-activated and ATB^0,+^-targeted LP containing sorafenib (inhibits xCT) and DOX (DS@MA-LS)**	Murine mammary carcinoma cells (in vitro and in vivo)	DS@MA-LS decreased tumour weight more effectively in vivo compared to free-drug treatment groups. Accumulation of LPs in the liver was not associated with liver injury.	Decrease in GPX4 protein and increase in MDA in vitro.	[156]
**LP rich in unsaturated lipids, containing iron and GSH-responsive DOX prodrug (LipoDSSSD)**	Murine melanoma	LipoDSSSD effectively killed tumour cells, but not healthy cells in vitro; more effective than other treatment groups at reducing tumour volume/weight in mice.	Increase in oxidised C11-BODIPY and depletion of GSH in vitro.	[157]
**Protoporphyrin IX (PpIX) sonosensitiser and ferumoxytol co-loaded LPs (Lipo-PpIX@Ferumoxytol)**	Murine mammary carcinoma (in vitro and in vivo)	Lipo-PpIX@Ferumoxytol induced dual ferroptotic and apoptotic cell death in cancer cell lines. The single components of the LP did not induce significant toxicity in cancer cells. Synergistic cell death was also observed in in vivo, as seen by reduced tumour weight and increased survival.	Increases in ROS were observed using DCFH-DA in vitro.	[158]
**Ultrasmall active catalysts (UACs, Cu*_x_*Fe_3–*x*_O_4_), camptothecin and DOX coated with pH-sensitive LP (LFCCD)**	Murine colorectal cancer (in vitro and in vivo)	Ferroptosis induced by UACs synergised with apoptosis induced by the two chemotherapeutic agents resulting in inhibited tumour growth in mice.	Increase in oxidised C11-BODIPY and depletion of GSH in vitro.	[159]
**CuO_2_ & artemisinin loaded LP (Lipo-ART@CPNs)**	Murine lung cancer	Lipo-ART@CPNs significantly reduced tumour burden compared to control and when combined with ultrasound irradiation and was the most effective treatment group.	Increase in oxidised C11-BODIPY in vitro. GPX4 protein downregulation in vitro and in vivo.	[160]
**PDGFB-targeted, iron-platinum nanoLP containing glucose oxidase (pLFePt-GOx)**	Triple-negative breast cancer	The pLFePt-GOx treatment group exhibited the lowest tumour volume in a mouse xenograft model.	Decreased GPX4 expression, increased ROS production in vitro.	[161]
**LPs loaded with nanoprobes and superparamagnetic iron oxide (L1/C-Lipo/DS), and LPs with GOx and DOX (L2/C-Lipo/GD)**	Murine mammary carcinoma	The combination of the two LPs had a great anti-tumour effect in a metastatic breast cancer mouse model compared to other treatment groups.	Increase in oxidised C11-BODIPY and cell death prevented by Fer-1 in vitro.	[162]
**Inhalable biomineralized LP loaded with DHA and pH-responsive calcium phosphate (LDM)**	Human lung cancer (in vitro and in vivo)	LDM induced ferroptosis and apoptosis, whereas empty LPs and DHA-only LPs did not. LDM suppressed tumour growth in vivo and did not induce histopathological changes in other organs.	Elevated ROS (DCFH-DA) and alleviation of cell death by Fer-1 and NAC in vitro. Reduced GPX4 expression in vitro and in vivo.	[163]
**GSH-triggered LPs rich in unsaturated soybean phospholipids encapsulating ferric ammonium citrate (FC-SPC-lipo)**	Murine mammary carcinoma (in vitro and in vivo)	FC-SPC-lipo induced significant lipid ROS and ferroptosis in 4T1 cells and accumulated in tumour tissues in vivo. FC-SPC-lipo reduced tumour volume compared to saline in vivo, whereas empty LPs, free FAC and saturated LPs (FC-HSPC-lipo) did not.	Increase in oxidised C11-BODIPY and decreased GPX4 expression in vitro and in vivo.	[164]
**PEGylated LPs loaded with a ferrocene, a Fenton catalyst (Fc-Lp-PEG)**	Murine mammary carcinoma (in vitro and in vivo) and glioma in vivo.	Fc-Lp-PEG induced lipid peroxidation and cytotoxicity in cancerous but not normal cell lines. Free Fc and Fc-Lp-PEG showed high tumour inhibition ratio (45.5% and 71%, respectively) and reduced tumour volume in vivo with minimal side effects.	Increased oxidised C11-BODIPY and MDA, decrease in GPX4 and GSH, and morphological changes consistent with ferroptosis by TEM in vitro. Increased MDA in vivo.	[165]
**LPs embedded with PEGylated 3 nm γ-Fe_2_O_3_ nanoparticles (Lp-IO)**	Murine mammary carcinoma (in vitro and in vivo)	Lp-IO induced significant lipid peroxidation and ferroptotic cell death in cancer cell lines, while LPs and iron oxide nanoparticles individually did not. Lp-IO inhibited tumour growth in a mouse model and synergised with DOX.	Increase in oxidised C11-BODIPY in vitro and in vivo. Cell death prevented by Fer-1 and Lip-1 in vitro.	[166]
**LP containing pH-triggered amphiphilic dendrimer releasing sorafenib and hemin (SH-AD-L)**	Human liver cancer (in vitro and in vivo)	SH-AD-L induced ferroptotic cell death in vitro with mild cytotoxic effects in normal liver cells. At high concentrations, S-AD-L (no hemin) induced a slight decrease in cell viability. SH-AD-L treatment decreased tumour growth in vivo.	Increase in oxidised C11-BODIPY and MDA, and cytotoxicity alleviated by Fer-1 in vitro.	[167]

ATB^0,+^, amino acid transporter B^0,+^; DOX, doxorubicin; FAC, ferric ammonium citrate; Fer-1, ferrostatin-1; GSH, glutathione; Lip-1, liproxstatin-1; lncRNA, long non-coding RNA; LP, liposome; MDA, malondialdehyde; MMP2, matrix metalloproteinase 2; MT1DP, metallothionein 1D pseudogene; NAC, N-acetyl-l-cysteine; NSCLC, non-small cell lung cancer PDGFB, platelet-derived growth factor subunit B; TEM, transmission electron microscopy; WT, wild type.

## Data Availability

Data sharing not applicable.

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
