# Peer review of "Ferroptosis in Haematological Malignancies and Associated Therapeutic Nanotechnologies"

_ijms, 2023, doi:10.3390/ijms24087661_

Round 1
Reviewer 1 Report
The review by Mynott et al., titled Nanotechnologies that Exploit Ferroptosis for the Treatment of Haematological Malignancies extensively describes the role and mechanisms of iron-dependent cell death, ferroptosis, in haematological malignancies. Different types of blood cancers and their ferroptosis-inducing therapies were very accurately described. A large amount of new literature has been reviewed.
Major comment:
The title of paper begins “Nanotechnologies”, however, only 7 and 8 chapters describe nanotechnologies, what is less than 24% of the text. The title of the paper could be clarified to avoid confusion.
Minor comments:
line 157 – different citation format – is it reference 53?
line 259 – sentence does not related to Figure 3, should it be Figure 2?
line 548 – it should be Figure 3.
Author Response
The title of paper begins “Nanotechnologies”, however, only 7 and 8 chapters describe nanotechnologies, what is less than 24% of the text. The title of the paper could be clarified to avoid confusion.
We have changed the title of the manuscript to “Ferroptosis in Haematological Malignancies and Associated Therapeutic Nanotechnologies”.
Minor comments:
line 157 – different citation format – is it reference 53?
Thank you for pointing this out. The correct numbered reference has been inserted in the text.
line 259 – sentence does not related to Figure 3, should it be Figure 2?
Thank you. Indeed it should be Figure 2 and this has been amended.
line 548 – it should be Figure 3.
Thank you. Indeed it should be Figure 3 and this has been amended.
Reviewer 2 Report
This is a very comprehensive and up to date review of ferroptosis and its possible clinical application in haematological malignancies, as well as how nanotechnologies can be incorporated to deliver such treatments. It was well written and the authors should be congratulated on constructing a document that was so easy to read despite the complex nature of the topic. My only comment would be to include the size of each of the nanoparticles illustrated in Figure 3. Other suggestions, if the authors choose, would be to comment on whether or not these nanotechnologies cross the blood brain barrier (important for treatment of CNS lymphoma), the disadvantages of each, and why liposomes seem to be the most popular for inducing ferroptosis.
Author Response
This is a very comprehensive and up to date review of ferroptosis and its possible clinical application in haematological malignancies, as well as how nanotechnologies can be incorporated to deliver such treatments. It was well written and the authors should be congratulated on constructing a document that was so easy to read despite the complex nature of the topic. My only comment would be to include the size of each of the nanoparticles illustrated in Figure 3. Other suggestions, if the authors choose, would be to comment on whether or not these nanotechnologies cross the blood brain barrier (important for treatment of CNS lymphoma), the disadvantages of each, and why liposomes seem to be the most popular for inducing ferroptosis.
The size ranges of the various nanotechnologies have been added to the legend of Figure 3 together with references.
With regard to the BBB, we have added a comment at the end of the first paragraph of section 7:
“An important property to consider when designing nanotechnologies, particularly for the treatment of central nervous system (CNS) lymphoma, is their ability to cross the blood brain barrier (BBB). This is a complex process and depends on nanoparticle size, shape and surface charge amongst other factors [141]. Overall, in terms of size, nanoparticles up to approximately 20 nm are large enough to avoid renal excretion while being small enough to penetrate the BBB. Moreover, those nanoparticles with a negative (or relatively lower) zeta potential show greater transport through the BBB [141].”
Due to the complex nature of the analysis, we have not discussed the disadvantages of each nanotechnology, however, we do mention why liposomes are popular for inducing ferroptosis in the last paragraph of section 7.